# A Comprehensive Eco-Driving Strategy for CAVs with Microscopic Traffic Simulation Testing Evaluation

**DOI:** 10.3390/s23208416

**Published:** 2023-10-12

**Authors:** Ozgenur Kavas-Torris, Levent Guvenc

**Affiliations:** Automated Driving Lab, Department of Mechanical and Aerospace Engineering, The Ohio State University, Columbus, OH 43210, USA; guvenc.1@osu.edu

**Keywords:** eco-driving, ecological cooperative adaptive cruise control, velocity trajectory, dynamic programming, traffic simulation

## Abstract

In this paper, a comprehensive deterministic Eco-Driving strategy for Connected and Autonomous Vehicles (CAVs) is presented. In this setup, multiple driving modes calculate speed profiles that are ideal for their own set of constraints simultaneously to save fuel as much as possible, while a High-Level (HL) controller ensures smooth and safe transitions between the driving modes for Eco-Driving. This Eco-Driving deterministic controller for an ego CAV was equipped with Vehicle-to-Infrastructure (V2I) and Vehicle-to-Vehicle (V2V) algorithms. This comprehensive Eco-Driving strategy and its individual components were tested by using simulations to quantify the fuel economy performance. Simulation results are used to show that the HL controller ensures significant fuel economy improvement as compared to baseline driving modes with no collisions between the ego CAV and traffic vehicles, while the driving mode of the ego CAV was set correctly under changing constraints. For the microscopic traffic simulations, a 6.41% fuel economy improvement was observed for the CAV that was controlled by this comprehensive Eco-Driving strategy.

## 1. Introduction

Fuel economy enhancement in road vehicles is a problem that researchers around the world have been working to improve for decades. Eco-Driving is a term used to describe the energy-efficient use of road vehicles. Some researchers have focused on improving the powertrain efficiency to improve the fuel economy in vehicles [1,2], whereas others have worked on utilizing Connected and Autonomous Vehicle (CAV) technologies for the same purpose [3,4,5]. Longitudinal autonomy and connectivity have also been utilized to achieve fuel savings for individual and platooning vehicles [6]. Robust control and model regulation were also used for vehicle control [7,8,9]. A parameter space with robustness was also utilized as another method for vehicle control [10,11,12,13], (p. 20 [14]), [15,16]. The problem being addressed here is how to improve the fuel efficiency of a vehicle (Eco-Driving) by using connectivity with the infrastructure and nearby vehicles. Existing solutions are first presented in the literature review below, followed by the contributions made in this paper. The aim of this paper is to improve the fuel economy, which is shown in the simulation experiment results parts of this paper.

Developments in Vehicle-to-Infrastructure (V2I) communication technology have enhanced the capabilities of CAVs. Researchers are able to study and enhance the fuel economy in vehicles by using vehicle connectivity technology in CAVs. CAVs can use roadway infrastructure information through V2I, where they receive information about traffic lights and STOP signs in order to reduce fuel consumption for conventional vehicles and battery power for electric vehicles. In V2I, CAVs receive traffic light and STOP sign locations, as well as the Signal Phase and Timing (SPaT) from traffic lights. Using this information, longitudinal control algorithms can be developed to modify the speed of the ego CAV in order to save fuel.

There is ample work in academia on algorithms utilizing V2I technology. Altan et al. developed a V2I algorithm and tested it at one signalized intersection to quantify how much fuel it saved for one connected vehicle [17]. Cantas et al. [18] and Kavas-Torris et al. [19] studied the fuel saving performance of the Pass-at-Green (PaG) V2I application in a microscopic traffic simulator through Monte Carlo simulations, as well as Hardware-in-the-loop (HIL) tests. Kavas-Torris et al. [20] analyzed the PaG V2I algorithm through microscopic traffic simulations, where varying but realistic traffic flows were present around the ego CAV. Sun et al. used a data-driven approach, where the optimal speed profile for a CAV thorough an intersection showed 40% fuel savings [21]. Asadi and Vahidi utilized V2I and a radar for a control algorithm that reduced the fuel consumption and idling time at traffic lights [22]. Li et al. also utilized SPaT to improve fuel savings [23]. Li et al. used V2I for Eco-Driving through Eco-Departure from a signalized intersection for CAVs with internal combustion engines [24].

Drivers interact with other drivers during daily driving activities and are bound by the speed of the slower preceding vehicle that they are following. To consider the Eco-Driving of a CAV in traffic, the preceding vehicle information also has to be taken into account by control algorithms in the ego CAV, such as the lead vehicle position and speed. Vehicles can also communicate with each other through V2V to obtain acceleration information and use it for fuel economy, emissions and safety benefits.

Cruise Control (CC) systems aim to keep the vehicle speed constant to aid the drivers on roadways and are particularly helpful for freeway driving [25]. CC designs usually employ classical control methods while approaches like fuzzy logic control have also been used [26]. They help in safety and are useful as Driver Assist Systems (DAS); however, CC models do not adjust the ego vehicle speed with respect to the outside input, such as the preceding vehicle’s position and speed.

Adaptive Cruise Control (ACC) has been widely used for saving fuel and improving safety for vehicles [27,28]. ACC is a valuable part of the Advanced Driver Assistance Systems (ADAS) and SAE Level 2 automated vehicles are equipped with ACC for car-following scenarios [29]. An ego vehicle equipped with a classical ACC uses cameras and radars to detect and track the preceding vehicle and actuators to control the ego vehicle speed [28]. Kural and Aksun Güvenç designed an ACC model by using Model Predictive Control (MPC) [30]. By reducing the unnecessary accelerations and decelerations as much as possible in the ego vehicle, ACC systems help to improve performance and indirectly save fuel. However, V2V technology is not utilized in ACC systems.

Cooperative Adaptive Cruise Control (CACC) enables V2V to be used for car-following scenarios [31]. In CACC, the ego vehicle receives information about the preceding vehicle from the preceding vehicle itself via V2V communication. Hu et al. utilized V2V technology for car following with an optimal look-ahead control framework for fuel savings [32]. Cantas et al. implemented a CACC algorithm, where the ego CAV received the acceleration of the preceding vehicle through V2V [33]. Kianfar et al. designed a CACC architecture that is capable of driving within a vehicle platoon while minimizing inter-vehicular spacing, attenuating shock waves and ensuring safety [34]. Rasool et al. used Pontryagin’s Minimum Principle (PMP) to improve fuel efficiency during car following with CACC [35]. Güvenç et al. designed and tested a CACC system for the Grand Cooperative Driving Challenge (GCDC) [36]. Naus et al. used the frequency-domain approach to design and experimentally validate a string-stable CACC system [37].

Ecological Cooperative Adaptive Cruise Control (Eco-CACC) is an improvement over the CACC system and aims to improve fuel efficiency by using road information while utilizing CACC in car-following scenarios or vehicle platoons. Zhai et al. designed an Eco-CACC model for a heterogeneous platoon with a time delay between the platoon agents [38]. Yang et al. modeled an Eco-CACC algorithm to compute the fuel-optimum vehicle trajectory through a signalized intersection that also handles queue effects [39]. Almannaa et al. designed an Eco-CACC model to reduce fuel consumption and achieve travel time savings around signalized intersections and also tested the system through field implementation [40].

There is also research on the energy management of vehicles using V2I and V2V in the recent literature. Zhang et al. focused on using a chaining neural network and an improved equivalent consumption minimization strategy (ECMS) with an equivalent factor (EF) to minimize energy consumption in a hybrid electric vehicle and showed a benefit ranging from 0.2% to 5% over the ECMS with a traditional adaptation law [41]. He et al. proposed an improved MPC-based strategy for energy management utilizing V2V and V2I for a hybrid vehicle [42]. Ma et al. used V2V for platooning and V2I for passing at intersections for a homogeneous platoon of connected electric vehicles [43].

In this study, a comprehensive Eco-Driving strategy was developed for a CAV equipped with V2I and V2V algorithms. The validation of the proposed strategy was carried out by using realistic simulations with other traffic generated by a microscopic traffic simulator. This study shows the relative fuel savings that each component provides to CAVs, how each component can be improved and what constitutes the largest effect on fuel savings. It has been shown that the complete Eco-Driving architecture presented in this paper is applicable to be used in real life in actual vehicles. The main contribution of this paper is the development and simulation validation of an integrated Eco-Driving system that uses V2I to handle realistic situations with infrastructure (STOP signs and traffic lights) and V2V to handle interactions with other vehicles. The other contributions that help this main contribution are summarized as follows:V2I and V2V algorithms were developed to control the longitudinal motion of a CAV for Eco-Driving.The High-Level (HL) controller was also tested in a traffic simulator with realistic traffic flow. The traffic vehicles were controlled by the traffic simulator and had default car-following models, which enabled them to change lanes when they were behind slower vehicles. Thus, the traffic vehicles created dynamically changing constraints on the HL controller. It was observed that the HL controller ensured that no collisions were observed between the ego CAV and traffic vehicles, and the driving mode of the ego CAV was set correctly under changing constraints.The High-Level (HL) controller designed for the comprehensive Eco-Driving of a CAV enabled fuel savings.

The rest of this paper is organized as follows: Section 2 describes the comprehensive Eco-Driving strategy for CAVs that was developed in this work. Section 3 details the deterministic High-Level controller. The microscopic traffic simulation environment is introduced in Section 4. Section 5 discusses the simulation results and comparative analysis based on various performance measures, followed by conclusions summarized in Section 6.

## 2. Complete Eco-Driving Strategy for a Connected and Automated Vehicle (CAV)

The schematic in Figure 1 displays a complete picture of the comprehensive Eco-Driving strategy for CAVs proposed in this paper. Firstly, the CAV needs to have a speed profile, which is called Eco-Cruise, that is route-dependent and fuel-optimal. This Eco-Cruise speed profile would assume normal operating conditions, meaning it would assume no surrounding traffic and infrastructure around the CAV. Additionally, the speed limit of the route and ride comfort with desired and safe acceleration and deceleration limits need to be enforced as constraints during the calculation of this fuel-optimal speed profile. This speed profile takes the route elevation into account, as well as the constraints of the vehicle, and can be calculated offline by using Dynamic Programming (DP). The Eco-Cruise mode shown in Figure 1 is the default driving mode, meaning that when the ego CAV does not interact with other vehicles or is not in the vicinity of traffic signs, Eco-Cruise is active to consume as little fuel as possible.

CAVs interact with roadway infrastructure, such as traffic lights and STOP signs. For the Eco-Driving of a CAV, when there is an upcoming traffic light and the traffic light Signal Phase and Timing (SPaT) information is broadcast from a Roadside Unit (RSU), then the ego CAV goes into Pass-at-Green (PaG) mode (the green-colored block in Figure 1). In this mode, the ego CAV picks up the traffic light state and duration, as well as the location, from the upcoming traffic light. Then, the V2I longitudinal control algorithm on the ego CAV makes a decision about either passing the traffic light or stopping for a red light. In order to pass the traffic light, the ego CAV can accelerate to a higher speed, keep its speed constant or decelerate to a lower speed. If one of these three states is possible, then the PaG calculates a smooth speed profile for the ego CAV to follow so that the fuel economy and ride comfort are maximized. For the state where the vehicle is not able to pass, the PaG calculates a smooth Eco-Approach to the traffic light so that the vehicle decelerates smoothly and spends as little time as possible while idling during the red light. Once the light turns green, the PaG calculates a smooth Eco-Departure speed profile from the traffic light.

For the Eco-Driving of a CAV, the ego CAV also interacts with STOP signs on roadways. STOP signs are usually not equipped with any type of V2I equipment; therefore, another tool needs to be used to obtain the STOP sign location information. In this architecture, the ego CAV is equipped with eHorizon (Autoliv Inc., Ogden, UT, USA, 2020), an electronic horizon that has a detailed map in it. Once the ego CAV gets close to the STOP sign location, it goes into Eco-Stop mode (the red-colored block in Figure 1). In Eco-Stop mode, using the STOP sign location information, an Eco-Approach speed profile is calculated that enables the ego CAV to decelerate smoothly in a fuel-optimal manner and stop at the STOP sign. After the ego CAV waits for 5 s at the STOP sign during the simulations, the Eco-Departure is subsequently activated to get the vehicle to speed up to the speed limit. A perception sensor like a camera and image processing should be used in conjunction with the electronic horizon map in practice to be certain of the STOP sign location and presence. While a 5 s wait period is fine for the fuel economy computations in this paper, the CAV should use perception and communication sensors to assess the safety of operation before proceeding to depart the STOP sign.

Other than the roadway infrastructure, CAVs also interact with other surrounding traffic agents. CAVs are equipped with perception sensors; hence, they can detect nearby objects or vehicles. For the Eco-Driving of a CAV, once the ego CAV detects a preceding vehicle, it needs to go into Eco-Cooperative Adaptive Cruise Control (Eco-CACC) mode (the light-orange-colored block in Figure 1). This mode uses V2V communication so that the ego CAV obtains the preceding vehicle’s information and uses that information to follow the preceding vehicle in a fuel-efficient manner.

When the preceding vehicle’s movement is too erratic or the preceding vehicle is moving too slowly, the ego CAV goes into Lane-Change mode (the gray-colored block in Figure 1). In Lane-Change mode, the ego CAV obtains the surrounding vehicles’ information, such as the vehicles’ speed and acceleration, as well as the vehicles’ position. Then, the model determines if it is safe to change lanes and executes lane changing. The main goal of a Lane Change in the Eco-Driving of a CAV is to make sure the ego CAV can maintain the optimal Eco-Cruise speed to obtain maximum fuel savings while also ensuring the safety of the ego vehicle and other nearby vehicles in adjacent lanes. If the leader vehicle changes lanes, it is not a leader vehicle anymore and the ego vehicle will revert back to Eco-Cruise until a new leader vehicle is encountered. If the Lane-Change mode commanded a Lane Change for the ego vehicle, but a new vehicle from adjacent traffic lanes joined the target lane, then the ego vehicle would either go back to the Eco-Cruise or car-following modes, depending on the speed of this new vehicle in front.

### 2.1. Fuel Optimization with Eco-Cruise

Dynamic Programming is a well-known solution that is used to find optimal benchmark solutions to various optimal control problems. Dynamic Programming (DP) [9] was used in the calculation of the fuel-optimal Eco-Cruise speed profile for a conventional vehicle. For Eco-Cruise, the problem was to minimize the road load acting on the vehicle (Figure 2) so that the fuel consumed by the vehicle would also be minimized.
(1)$$Road\;Load=F_{rolling}+F_{aero}+F_{grade}$$
(2)$$Road\;Load=mgr_{0}\cos\left(\alpha \left(s\right)\right)+\left(\frac{1}{2}\rho_{air}A_{f}C_{D}v^{2}\right)+mg\sin\left(\alpha \left(s\right)\right)$$

The road load (Figure 2) equation given in (1) has three parts. The first part is the rolling resistance $F_{rolling}$, and this term depends on the tire properties, vehicle speed and road conditions. In Equation (2), m is the vehicle mass with the rotating inertia factor, $r_{0}$ is a parameter of the rolling resistance equation and α is the road grade. The second term is $F_{aero}$ and refers to the aerodynamic drag term. $F_{aero}$ depends on the vehicle speed and frontal cross-section area of the vehicle. In Equation (2), $\rho_{air}$ is the density of air, $A_{f}$ is the front cross-sectional area, $C_{D}$ is the drag coefficient and v is the vehicle speed. $F_{grade}$ is the road grade term, and it depends on the vehicle mass and the road grade.

The power that needs to be provided from the engine in a vehicle to beat road load and enable acceleration can be expressed as follows:(3)$$P=F_{x}v=\left(m_{e}\frac{dv}{dt}+\frac{1}{2}\rho_{air}\cdot A_{f}\cdot C_{D}\cdot v^{2}+m\cdot g\cdot r_{0}\cdot \cos\left(\alpha \right)+m\cdot g\cdot \sin\left(\alpha \right)\right)v$$
where P is the power, $F_{x}$ is the force required at the tires and $m_{e}\frac{dv}{dt}$ is the force required to accelerate. The rest of the terms in Equation (3) come from the road load acting on the vehicle, which was given in Equation (2). Using the power expression given in Equation (3), the fuel rate that is consumed by the vehicle when it is traveling can be expressed as follows:(4)m˙f=Pηt+Paccessoriesηe
where ${\dot{m}}_{f}$ is the fuel rate, $P_{accessories}$ is the power required to keep the accessories running, $\eta_{t}$ is the transmission efficiency and $\eta_{e}$ is the engine efficiency. This expression for the fuel rate given in Equation (4) can be used as the cost function that needs to be minimized for this analysis. Further details for this optimal control formulation can be found in [44].

In this paper, the fuel-optimal DP solution presented here was used for different driving modes. Firstly, the driving mode called Eco-Cruise, where the fuel-optimal speed profile is calculated offline by using road information, was found by using DP. Additionally, the Eco-Stop mode, where the ego vehicle approaches a STOP sign fuel-economically, also utilized DP. Finally, the Eco-Departure mode, where the ego vehicle departs from a traffic light or STOP sign, also used DP. These solutions were all distance-based solutions, as presented earlier.

In the DP solution, the whole trip horizon is divided into segments. Additionally, the solution space is also divided into nodes. The solution starts from the end point, where the desired vehicle speed and vehicle location are known. The cost in terms of the fuel rate seen in Equation (4) is assigned to each link to move from the current node to each previous neighboring node in backward propagation. Then, the feasibility constraint of going from one node to the next is checked, where the acceleration and deceleration, as well as the jerk-rate limits, are enforced. Details about this approach can be found in [44].

### 2.2. Vehicle-to-Infrastructure (V2I) Interactions of a CAV

A vehicle traveling from a starting location to a traffic light (or a STOP sign) can be seen in Figure 3. In Figure 3, $x_{ego}$ is the position, $v_{ego}$ is the speed and $a_{ego}$ is the acceleration of the ego vehicle. $TL_{location}$ is the traffic light location, $TL_{SPaT}$ is the traffic light state and duration and $STOP_{location}$ is the location of the STOP sign.

In this paper, when it comes to V2I communication, the aim is to design control algorithms that minimize the fuel consumption in a vehicle. Fuel consumption can be reduced through the utilization of V2I so that the vehicle control algorithms can obtain roadway infrastructure information and use it to consume less fuel. The optimal control problem can be defined with the objective function (5):(5)$$\underset{T_{e}\left(t\right),\;F_{b}\left(t\right)}{\underbrace{minimize}}\;\;J\left(u\left(t\right)\right)=L_{N}\left(s\left(t_{f}\right),v\left(t_{f}\right)\right)+\int_{0}^{t_{f}}L_{k}\left(s\left(t\right),v\left(t\right),T_{e}\left(t\right),F_{b}\left(t\right),t\right)dt$$
where
$L_{k}$ is the running cost and $L_{N}$ is the terminal cost. Additionally, u(t) is the input, s(t) is the distance, $t_{f}$ is the final time, v(t) is the vehicle velocity, $T_{e}\left(t\right)$ is the engine torque and $F_{b}\left(t\right)$ is the brake force. The states are subject to
(6)$$\frac{ds\left(t\right)}{dt}=v\left(t\right)$$
(7)$$\frac{dv\left(t\right)}{dt}=K_{T_{e}}T_{e}-K_{F_{b}}F_{b}-gr_{0}\cos\left(\alpha \left(t\right)\right)-\frac{1}{2m}\rho_{air}A_{f}C_{D}v{\left(t\right)}^{2}-g\sin\left(\alpha \left(t\right)\right)$$
where Equation (6) expresses that the derivative of the position is equal to the speed. In Equation (7), m is the vehicle mass, $r_{0}$ is a parameter of the rolling resistance equation, α(t) is the road grade, $\rho_{air}$ is the density of air, $A_{f}$ is the front cross-sectional area, $C_{D}$ is the drag coefficient, v(t) is the vehicle speed and g is the gravitational acceleration. The vehicle model expression given in Equation (7) shows that road load and brake force subtracted from the total powertrain force to the wheels is equal to the vehicle acceleration. There are initial and final algebraic constraints on the states of the position and speed, and they are as follows:(8)$$s\left(0\right)=s_{initial}=0$$
(9)$$s\left(t_{f}\right)=s_{final}=s_{f}$$
(10)$$v\left(0\right)=v_{initial}=v_{i}=0$$
(11)$$v\left(t_{f}\right)=v_{final}=v_{f}=0$$
(12)$$v_{min}\left(t,s\left(t\right)\right)\leq v\left(t\right)\leq v_{max}\left(t,s\left(t\right)\right)$$
where $s_{initial}$ (8) is the initial vehicle position and $s_{f}$ (9) is the final vehicle position. Additionally, $v_{i}$ (10) is the initial vehicle speed, $v_{f}$ (11) is the final vehicle speed and $v_{min}$ (12) is the minimum allowable speed for the vehicle. The speed limit of the roadway is enforced as the $v_{max}$ (12) constraint, which is the maximum allowable speed of the vehicle. There are also algebraic constraints on the input’s engine torque $T_{e}$ (13) and brake force $F_{b}$ (14):(13)$$T_{e,min}\left(v\left(t\right),t\right)\leq T_{e}\left(t\right)\leq T_{e,max}\left(v\left(t\right),t\right)$$
(14)$$0\leq F_{b}\left(t\right)\leq F_{b,max}\left(v\left(t\right),t\right)$$

The optimal control problem posed here was solved by using Dynamic Programming for the case where the ego CAV approaches a STOP sign. Different Eco-Approach profiles were calculated for approaching the STOP sign, and depending on the instantaneous speed of the ego CAV when it was within 300 m of the STOP sign, the appropriate profile was chosen during the simulations.

For the interactions between the ego CAV and traffic lights, Pass-at-Green (PaG) was used. PaG is a V2I application that uses roadway infrastructure information to eliminate or decrease idling at red lights to decrease the fuel consumption for the ego vehicle. PaG operates under deterministic control by using the input, which includes the distance to the upcoming traffic light, Signal Phase and Timing (SPaT) information received from the upcoming traffic light, instantaneous actual vehicle speed, maximum acceleration and maximum deceleration limits and jerk limit for ride comfort. Using these inputs, the PaG calculates a smooth and fuel-economic speed profile so that the vehicle can pass the upcoming traffic light.

Depending on the distance to the traffic light and the SPaT information from the upcoming traffic light, the PaG chooses one of four options for the recommended vehicle-speed trajectory. These PaG states are as follows:*Cruise State:* the vehicle keeps its speed constant and passes the traffic light when the light is green.*Increase Speed State:* The vehicle accelerates to a higher speed, travels at a constant speed when it is passing the green traffic light and then decelerates to the initial lower speed. The vehicle obeys speed limits, as well as acceleration, deceleration and jerk limits for ride comfort.*Eco-Approach State:* The vehicle cannot catch the current green light; therefore, it decelerates to a stop at the traffic light. Then, after the traffic light turns green, the vehicle smoothly accelerates to a higher speed and passes the traffic light. The vehicle obeys speed limits, as well as acceleration and deceleration limits for ride comfort.*Decrease Speed State:* The vehicle decelerates to a lower speed, travels at a constant speed when it is passing the traffic light and then accelerates to the initial higher speed. The vehicle obeys speed limits as well as acceleration and deceleration limits for ride comfort.

More information on the PaG can be found in [17,18,19,44].

### 2.3. Vehicle-to-Vehicle (V2V) Interactions of a CAV

An ego CAV following a lead connected vehicle can be seen in Figure 4. $x_{ego}$ and $x_{lead}$ are the positions of the ego and lead vehicles, respectively. ${\dot{x}}_{ego}$ and ${\dot{x}}_{lead}$ are the speeds of the ego and lead vehicle, respectively. ${\ddot{x}}_{ego}$ and ${\ddot{x}}_{lead}$ are the accelerations of the ego and lead vehicle, respectively. It should be noted that the sinusoidal-looking perturbation in the speed profile of Figure 4 is for illustration purposes only and represents a perturbation (not necessarily sinusoidal) that the ego vehicle does not want to follow.

Fuel consumption in CAVs can be reduced by the utilization of V2V so that the vehicle control algorithms can obtain the lead vehicle’s information and use it to consume less fuel.

In order to prevent a collision from happening between the lead vehicle and the ego CAV, the following algebraic constraint also needs to be enforced. These constraints are as follows:(15)$$x_{actual}=x_{lead}-x_{ego},\;x_{actual}>0$$
where $x_{actual}$ (15) is the actual distance between the lead and the ego vehicle, and it needs to always be larger than zero to ensure that the vehicles do not collide.

ACC, CACC and Eco-CACC models with Proportional-Derivative (PD) feedback control and a constant time-gap spacing policy were designed in order for the ego CAV to safely follow the lead vehicle. Eco-CACC used a preceding acceleration feedforward compensator that filtered high-frequency acceleration disturbances of the preceding vehicle. More information about the V2V models can be found in [45,46].

## 3. The High-Level Controllers for V2I, V2V and V2I + V2V

In this section, the deterministic control algorithms that were developed for the Eco-Driving of a CAV are explored further.

### 3.1. High-Level (HL) Controller for V2I with No Traffic

The High-Level (HL) controller for V2I with no traffic handles how the ego CAV behaves when it is traveling on a roadway with no other vehicle around it and is implemented as a state-flow chart. The aim is to determine when the CAV has to switch between the different driving modes of the Eco-Driving of the CAV architecture presented in Figure 1. This controller ensures the seamless transition from one driving mode to the next.

Depending on deterministic conditions, such as the current upcoming traffic light state and duration, the distance between the infrastructure elements (the traffic lights and STOP signs) and the ego vehicle, as well as the instantaneous vehicle speed, the controller is tasked to make a decision to switch between driving modes. The flow chart for the deterministic control algorithm for the fuel-economic Eco-Driving of a single CAV with no traffic can be seen in Figure 5.

As seen in Figure 5, the ego CAV aims to maintain its speed as close to the Eco-Cruise speed as possible. The Eco-Cruise speed is the fuel-economic speed profile that is route-dependent and is calculated offline prior to the trip. In case there is an upcoming traffic light, the Pass-at-Green (PaG) V2I algorithm takes over control of the ego vehicle. If there is a STOP sign, then Eco-Stop mode is activated to make the vehicle stop smoothly at the sign. After stopping at the STOP sign for a few seconds, Eco-Departure takes over and makes the ego vehicle accelerate smoothly. The HL controller makes sure the correct driving mode is active and mode transitions are smooth to save as much fuel as possible.

### 3.2. High-Level (HL) Controller for V2V with Traffic

This High-Level (HL) controller for V2V with traffic aims to make transitions between driving modes correctly and smoothly so that the ego vehicle speed does not jump abruptly when the driving mode changes. The flowchart for the controller is seen in Figure 6, where the Eco-Cruise speed is the fuel-economic and road-dependent speed profile for the ego vehicle to follow to consume less fuel. When there is a preceding vehicle with no V2V communication, the ACC model is activated and the ego CAV safely follows the lead vehicle. If the preceding vehicle is equipped with V2V and does not have an erratic driver, then the CACC takes over and follows the lead vehicle smoothly while keeping a safe distance between vehicles to prevent a collision. If the preceding vehicle with V2V has an erratic driver, then Ecological Cooperative Adaptive Cruise Control (Eco-CACC) takes over control to follow the erratic leader without responding to its high-frequency accelerations in order to maintain fuel savings and safety. If the leader is erratic and lane changing is possible for the ego vehicle, then the ego vehicle changes its lane and maintains the Eco-Cruise speed.

The driving modes shown in Figure 6 have different controllers, and when they all run simultaneously during testing, the recommended vehicle speeds from each driving mode are usually different. If driving modes were to switch immediately with no transition, then the recommended speeds would not be continuous and cause the actual ego vehicle speed to jump abruptly. To overcome this problem, a Transition State is added to smoothly transition between driving modes. The algebraic equation for the Transition State to smoothly increase the vehicle speed is as follows:(16)$$v_{trans}=v_{trig}+v_{chg,acc}$$
(17)$$v_{chg,acc}=\left(v_{lim}-v_{trig}\right)\left({\left(\frac{t_{act}-t_{trig}}{4\left(v_{lim}-v_{trig}\right)}-1\right)}^{3}+1\right)\;$$
where $v_{trans}$ (16) is the recommended transition speed for the vehicle, $v_{trig}$ is the vehicle speed when the driving-mode transition started and $v_{chg,acc}$ is the speed change needed for the ego vehicle to travel at the higher speed limit. In Equation (17), $v_{lim}$ is the actual speed limit of the road, $t_{act}$ is the actual simulation time and $t_{trig}$ is the time instant when the driving-mode transition starts. The third-order power equation that comprises the variables seen in Equation (17) ensures that the recommended speed is smooth when driving modes are switched and the ego CAV accelerates.

When the Eco-Cruise speed is smaller than the instantaneous vehicle speed, the following Equation (18) ensures that the vehicle decelerates slowly. In Equation (18), $v_{chg,dec}$ is the speed change needed for the ego vehicle to travel at the lower speed limit. In Equation (19), $v_{lim,low}$ is the user-set lower speed limit, $t_{act}$ is the actual simulation time and $t_{trig}$ is the time instant when the driving-mode transition starts. A third-order power equation that comprises the variables seen in Equation (19) ensures that the recommended speed is smooth when driving modes are switched and the ego CAV decelerates. When the Eco-Cruise speed catches up to the vehicle speed, then the recommended speed for the CAV to follow switches back to the Eco-Cruise speed:(18)$$v_{trans}=v_{trig}-v_{chg,\;dec}$$
(19)$$v_{chg,dcc}=\left(v_{lim,low}-v_{trig}\right)\left({\left(\frac{t_{act}-t_{trig}}{4\left(v_{lim,low}-v_{trig}\right)}-1\right)}^{3}+1\right)\;$$

### 3.3. High-Level (HL) Controller for V2V and V2I with Traffic

The HL controller for V2V and V2I with traffic was designed as a state-flow diagram in Simulink, and the flowchart for the HL controller decision-making process can be seen in Figure 7. The default mode is the Eco-Cruise mode, where the precalculated fuel-economic DP profile is the desired speed profile for the vehicle. The Eco-Cruise speed profile also makes sure the ego vehicle drives in a fuel-economic manner around STOP signs. When there is a lead vehicle in close proximity to the ego vehicle, car-following models are activated to safely and closely follow the preceding vehicle. When there is a traffic light ahead, the mode is switched to the PaG V2I algorithm. After the ego vehicle passes the traffic light, depending on the instantaneous speed of the vehicle, the transition modes are activated (speed up or speed down).

## 4. Microscopic Traffic Simulation Environment

A simulation environment was set up by using Simulink and the Vissim commercial traffic simulator to run co-simulations by using the COM interface capability of Vissim [19,47]. Details about setting up a COM connection between Simulink and Vissim can be found in [47]. Other than the COM interface between Simulink and Vissim, there was no specific Matlab Simulink package that was installed for the simulation experiments. During the co-simulations, realistic traffic information was being sent from Vissim to Simulink. The ego vehicle with a mid-sized vehicle powertrain was being controlled by the High-Level (HL) controller in Simulink. The fuel consumption model was also in Simulink, and the realistic fuel consumption values were achieved by using multi-dimensional tables that replicated the behavior of a real vehicle engine. The HL controller determined which action to take and which driving mode to activate in response to the realistic traffic and infrastructure information received from the traffic simulator.

The simulation environment designed in Vissim is called the Arlington Route and it has one STOP sign, five traffic lights and is 6873 m long. The Arlington Route can be seen below in Figure 8.

The speed limit, traffic sign locations and route-dependent fuel-economic DP solution for the Eco-Cruise driving mode for the Arlington Route can be seen below in Figure 9.

The pink ego vehicle approaching a traffic light at an intersection with other traffic vehicles around it during the traffic simulation can be seen in Figure 10. During the simulation, the ego vehicle was controlled by the HL controller to save fuel by smoothly approaching traffic lights and STOP signs. At the same time, whenever there was a vehicle in front of the ego vehicle and the distance between the ego and lead vehicles was less than 50 m, ACC, CACC or Eco-CACC were activated to prevent collisions between the vehicles during car following.

The traffic-vehicle compositions were the same at each simulation. Additionally, the traffic simulator spawned vehicles at a common start time for each simulation, meaning that the vehicle with a specific ID entered the roadway at the same timestamp across all simulation cases. This unity ensures that the simulation results can be compared with each other since the traffic vehicles that interact with the ego vehicle appear in the simulator at the same time. Additionally, the traffic light periods for each traffic light were the same across all simulations. When it comes to experimental parameters, the distance traveled by vehicles, the inter-vehicular distance between the ego and leader vehicle, vehicle speed, simulation time, distance to traffic lights and STOP signs, SPAT for traffic lights and HL controller state were recorded and analyzed for system performance.

Depending on the test case and whether there were other traffic vehicles around the ego vehicle for V2V, or V2I communication with the road infrastructure, one of the three HL controllers presented in Section 3 was used.

## 5. Results and Discussion

To assess the fuel economy performance of the V2I and V2V algorithms in a traffic network, five different simulations were run. For case 1, the ego vehicle was commanded to follow the fuel-economic DP profile in Eco-Cruise mode with no other traffic vehicles around in the simulation. 

For the second case, the ego vehicle was commanded to follow the same Eco-Cruise speed as the first case while also interacting with STOP signs by using Eco-Stop and traffic lights by using PaG. For case 2, the HL controller for V2I with no traffic presented in Section 3.1 was utilized during the simulations.

The third simulation case built on top of the second simulation case, where Eco-Cruise, Eco-Stop and PaG were all working in tandem, and there were also traffic vehicles around the ego vehicle. Whenever the ego vehicle was in the vicinity of a lead vehicle, the ACC mode was activated. The fourth simulation case used the same V2I models, and when there was a lead vehicle ahead, the CACC mode was activated. The fifth and final simulation case used the same V2I models as the fourth case, except the car-following model that was used when there was a lead vehicle in front of the ego vehicle for this case was Eco-CACC. For cases 3, 4 and 5, the HL controller for V2V and V2I with traffic, which was presented in Section 3.3, was used.

The speed profile for the ego vehicle when there were no other traffic vehicles around can be seen in Figure 11. The light blue line represents the ego vehicle speed when it was commanded to follow the DP offline-calculated Eco-Cruise profile in Figure 9. The light red line represents the vehicle speed when the vehicle was around a traffic light, and the SPaT information was used to modify the speed profile for case 2.

The results of the third simulation case, where there were other traffic vehicles in the traffic simulator and the ego vehicle was equipped with the V2I algorithms and ACC, can be seen in Figure 12. Whenever the distance between the ego vehicle and the lead vehicle was below 50 m, the ACC took over control to make sure no collision could occur. If the distance between the ego and the lead vehicle was larger than 50 m, the HL controller commanded the ego vehicle to either follow the Eco-Cruise trajectory or the PaG trajectory to save fuel. During this simulation, around 500 s, the PaG commanded the vehicle to accelerate to pass the traffic light, which was not observed in cases 4 and 5. This acceleration-to-pass behavior observed in case 3 resulted in the ego vehicle having the highest fuel consumption among cases 3, 4 and 5.

The results of the fourth simulation case, where there were other traffic vehicles in the traffic simulator and the ego vehicle was equipped with the V2I algorithms and CACC, can be seen in Figure 13. The HL controller handled having a preceding vehicle ahead of the ego vehicle the same as the ACC case. Towards the end of the simulation in case 4, the ego vehicle switched into car-following mode with CACC. In CACC mode, the ego vehicle tried to follow the lead vehicle at a safe distance. During the simulation, the lead vehicle was driving faster than the ego vehicle, which resulted in the ego vehicle accelerating to a higher speed to keep up with the lead vehicle around 730 s. This resulted in the ego vehicle having a higher fuel consumption in case 4 (Figure 13), where the ego vehicle used CACC compared to case 5 (Figure 14), where the ego vehicle used Eco-CACC for car following.

The results of the fifth and the final simulation case, where there were other traffic vehicles in the traffic simulator and the ego vehicle was equipped with the V2I algorithms and Eco-CACC, can be seen in Figure 14. Similar to the previous cases with ACC and CACC, the HL controller handled the state transitions.

The fuel consumed by the ego vehicle in each of the two simulation cases, where there was no other traffic flow around the ego vehicle, was recorded, and the percentage of the fuel consumption reduction in the models was calculated with respect to the simulation case 1 (Table 1). During case 1 and case 2, there were no other vehicles around the ego vehicle to interact with by using V2V. When the ego vehicle could use V2I in case 2, the fuel consumed by the ego vehicle decreased compared to using the Eco-Cruise-only simulation in case 1, where the ego vehicle stops at all traffic lights and STOP signs.

Three simulations were run, where there was another vehicle around the ego vehicle, and the results are summarized in Table 2. Traffic vehicles that constrained the motion of the ego vehicle were present for cases 3, 4 and 5. Compared to ACC for car following in case 3, using CACC in case 4 resulted in a 1.51% fuel economy improvement. Moreover, using the Eco-CACC in case 5 was even more beneficial in reducing the fuel consumed by the ego vehicle. The fuel consumption decreased by 6.41% when using Eco-CACC in case 5 compared to using ACC in case 3.

## 6. Conclusions and Future Work

In this paper, a comprehensive Eco-Driving strategy with V2I and V2V algorithms was tested in a realistic microscopic traffic simulation environment, where a real-life route in Columbus, Ohio, USA, was modeled in a traffic simulator with the same number of lanes, speed limits, traffic lights and STOP signs. When PaG was active and used traffic infrastructure information in case 2, 3.46% less fuel was consumed compared to only using the Eco-Cruise speed profile case 1. For the simulation cases that required car following, it was shown that using CACC and Eco-CACC with V2V was more beneficial than using only ACC. The ego vehicle consumed 1.51% and 6.41% less fuel as compared to ACC only (case 3) for car following when CACC (case 4) and Eco-CACC (case 5) were used, respectively. Moreover, it was seen that Eco-CACC, which was modeled with a filter to attenuate the acceleration of the lead vehicle, consumed less fuel than CACC, which used the lead vehicle acceleration without filtering it.

For future work, the different driving modes that were presented here can be combined as part of an MPC with varying constraints under different driving conditions to improve the complete Eco-Driving strategy of the CAV presented in this paper.

There is also potential for improvement for the High-Level (HL) controller. In the simulation results, it was seen that for some cases during car following, the HL controller switched between different driving modes very rapidly. In real-life implementations, this rapid switching between driving modes would diminish the ride comfort for the passengers. To eliminate this rapid switching issue in the HL controller, a dead zone can be included in the controller. When controllers have dead zones, they do not respond to the change in the input within the dead zone region [48]. By exploring the addition of a dead zone to the HL controller, the rapid switching issue might be eliminated. 

Within the scope of this paper, it was assumed that the functional safety of the ego CAV was satisfied and there were no malicious agents for the V2I, V2V and V2X communication. However, in real life, there could be cyber-security threats to the functional safety of a CAV due to malicious road agents. For example, malicious agents could broadcast inaccurate acceleration information to other CAVs on the roadway, or they could intentionally drive in an erratic manner. For safe and reliable real-life implementation and VIL testing, the cyber-security and functional safety aspects of CAVs need to be explored further. 

To obtain real-world behavior when the CAVs are deployed, datasets dedicated to CAVs are needed. These were not created in the current paper, but there are such papers in the literature where such data are collected. For example, the paper in [49] presented a dedicated dataset for analyzing CAVs’ behavior.

## Figures and Tables

**Figure 1 sensors-23-08416-f001:**
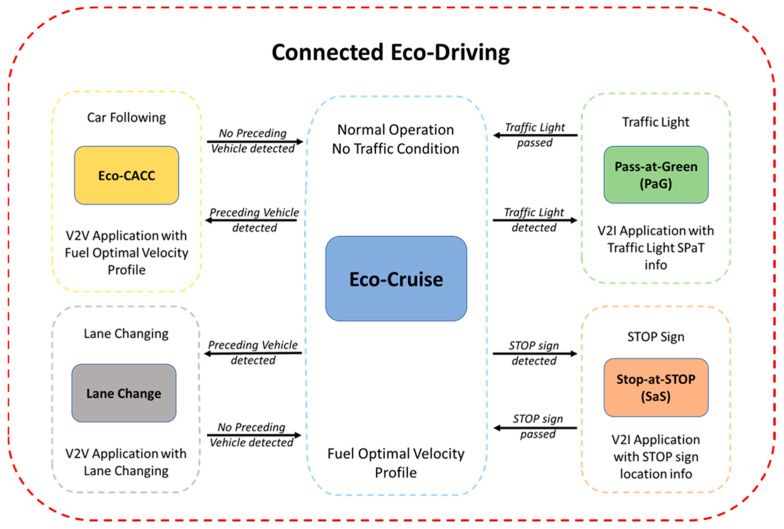
Comprehensive Eco-Driving architecture of CAVs.

**Figure 2 sensors-23-08416-f002:**
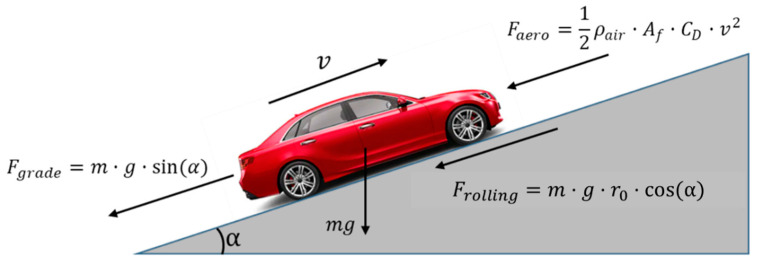
Road forces acting on a vehicle.

**Figure 3 sensors-23-08416-f003:**
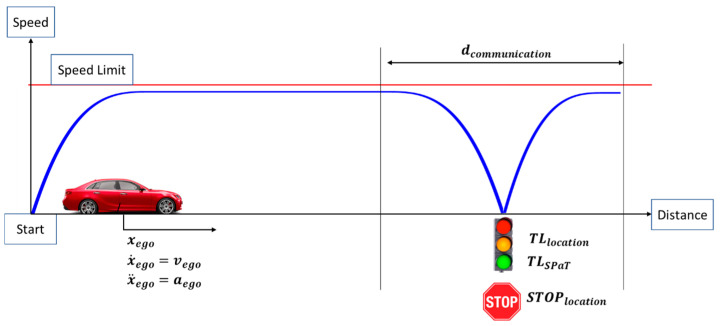
V2I interaction as an optimal control problem.

**Figure 4 sensors-23-08416-f004:**
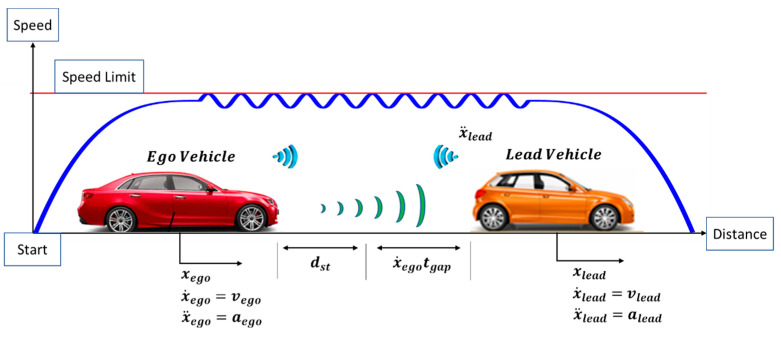
Car following a CAV as an optimal control problem.

**Figure 5 sensors-23-08416-f005:**
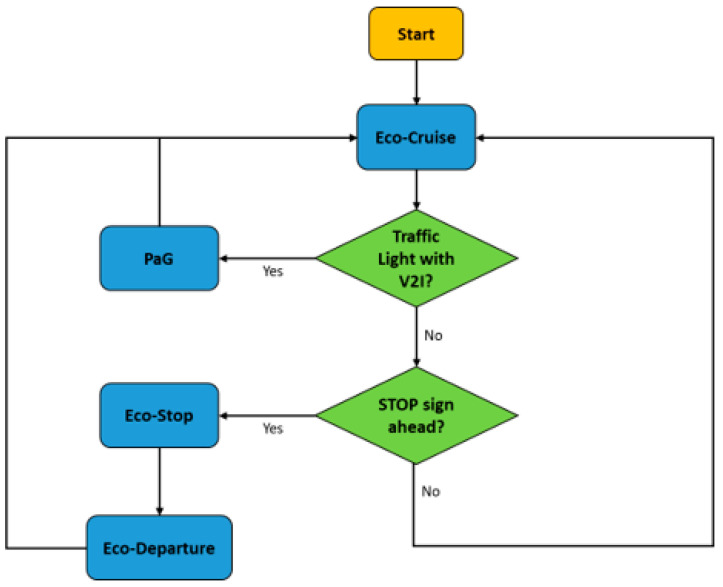
Flowchart for the HL controller with no traffic for V2I.

**Figure 6 sensors-23-08416-f006:**
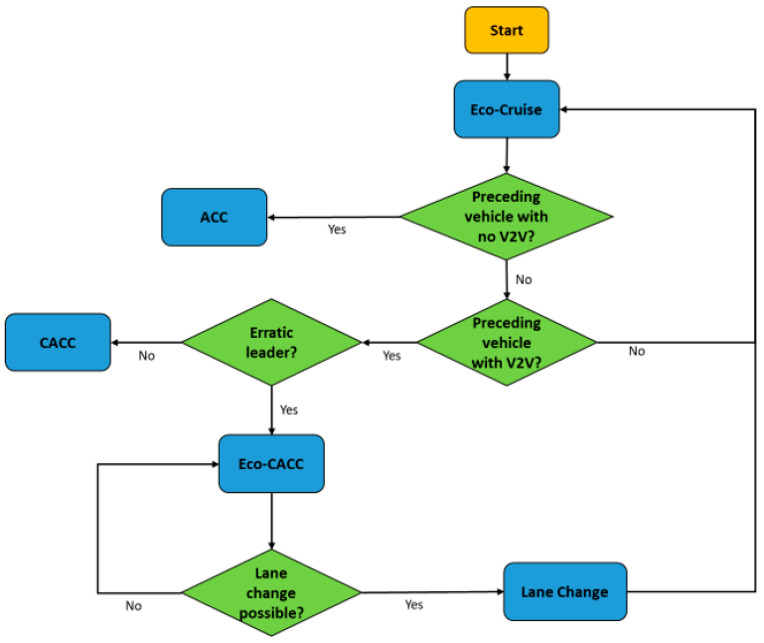
Flowchart for the HL controller with traffic for V2V.

**Figure 7 sensors-23-08416-f007:**
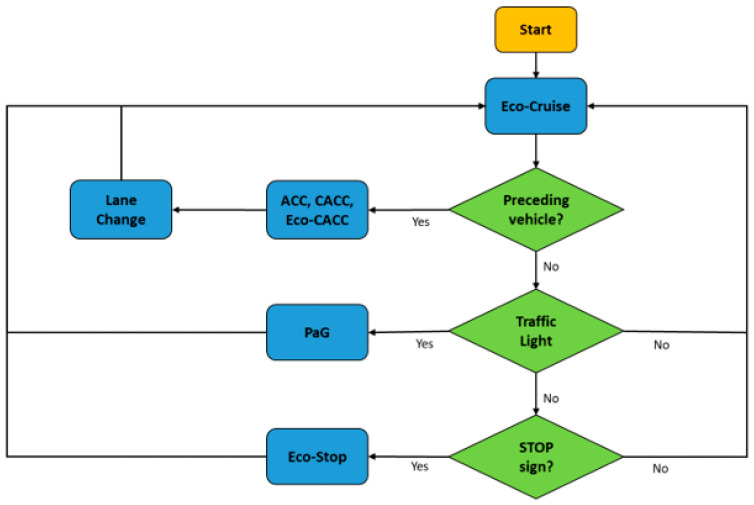
Flowchart for the HL controller with traffic for V2V and V2I.

**Figure 8 sensors-23-08416-f008:**
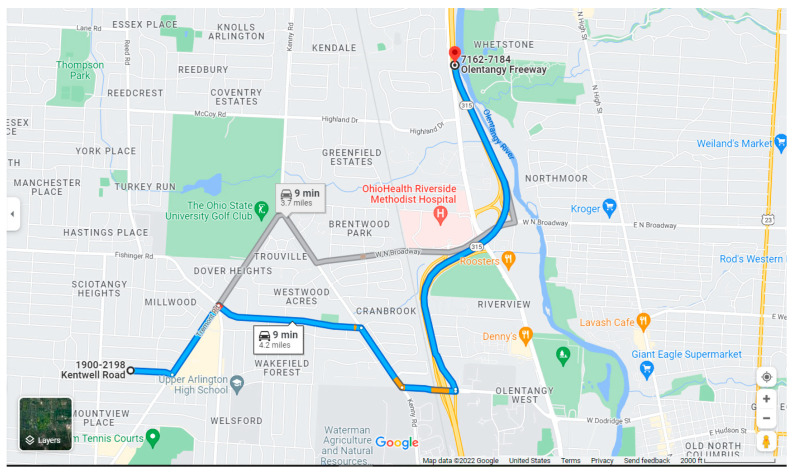
Arlington Route from “Google Maps” (2022).

**Figure 9 sensors-23-08416-f009:**
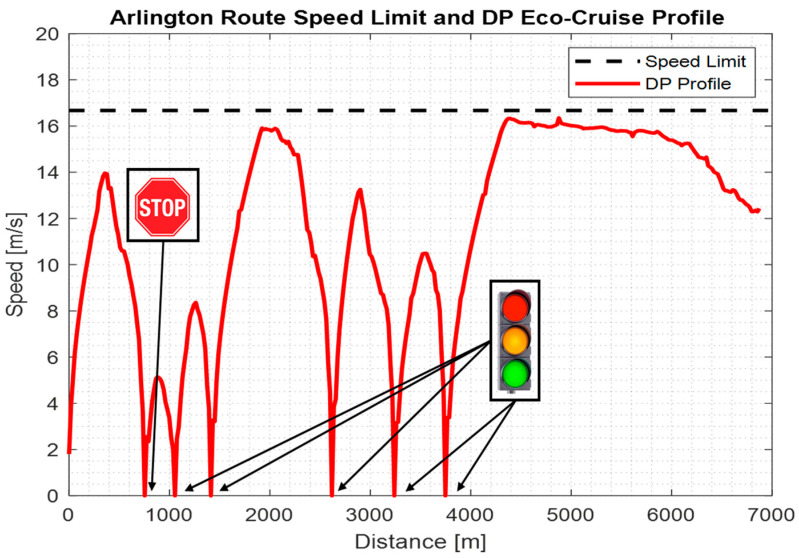
Characteristics of the Arlington Route.

**Figure 10 sensors-23-08416-f010:**
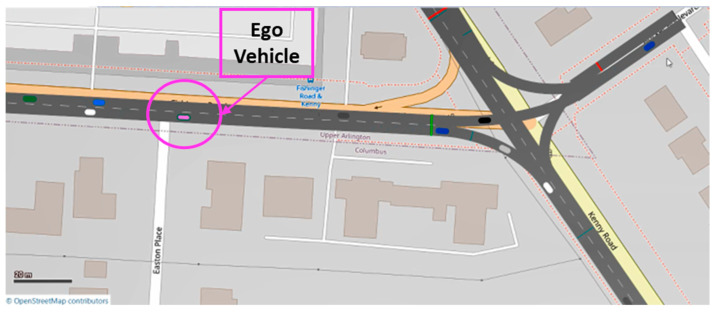
Ego vehicle approaching an intersection in the traffic simulation.

**Figure 11 sensors-23-08416-f011:**
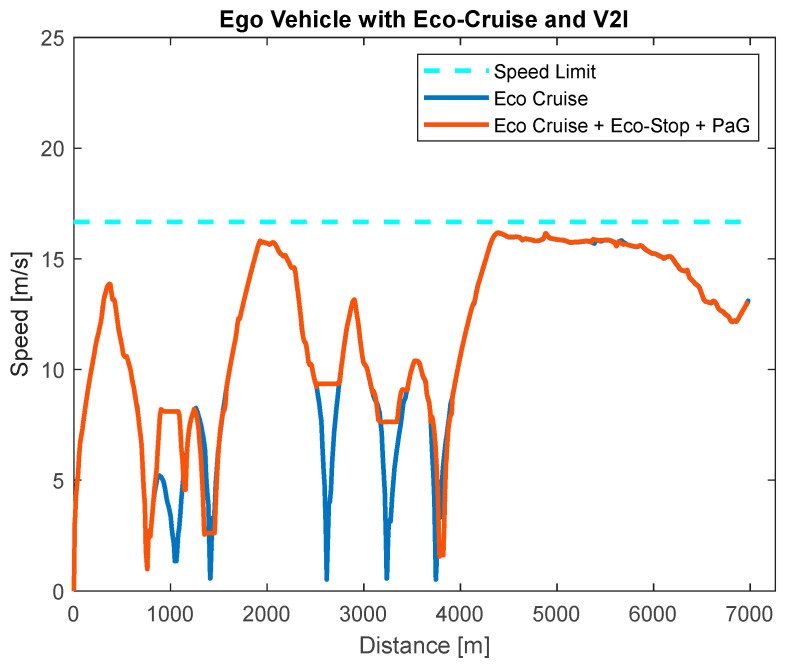
Ego vehicle with Eco-Cruise only (case 1) and Eco-Cruise + Eco-Stop + PaG (case 2).

**Figure 12 sensors-23-08416-f012:**
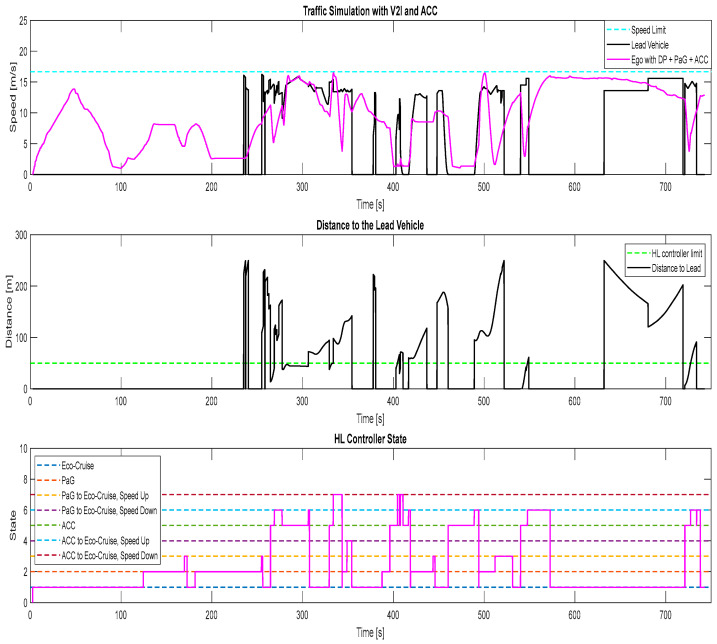
Traffic simulation for ego vehicle with V2I and ACC, case 3.

**Figure 13 sensors-23-08416-f013:**
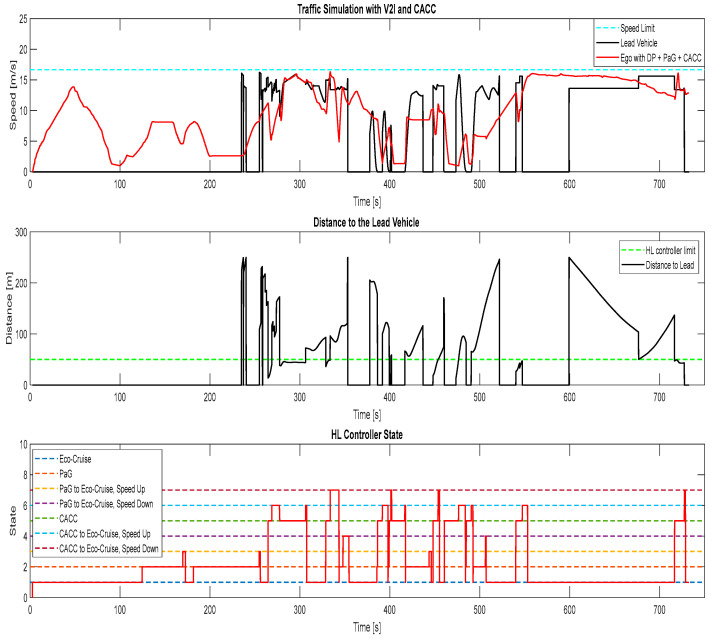
Traffic simulation for ego vehicle with V2I and CACC, case 4.

**Figure 14 sensors-23-08416-f014:**
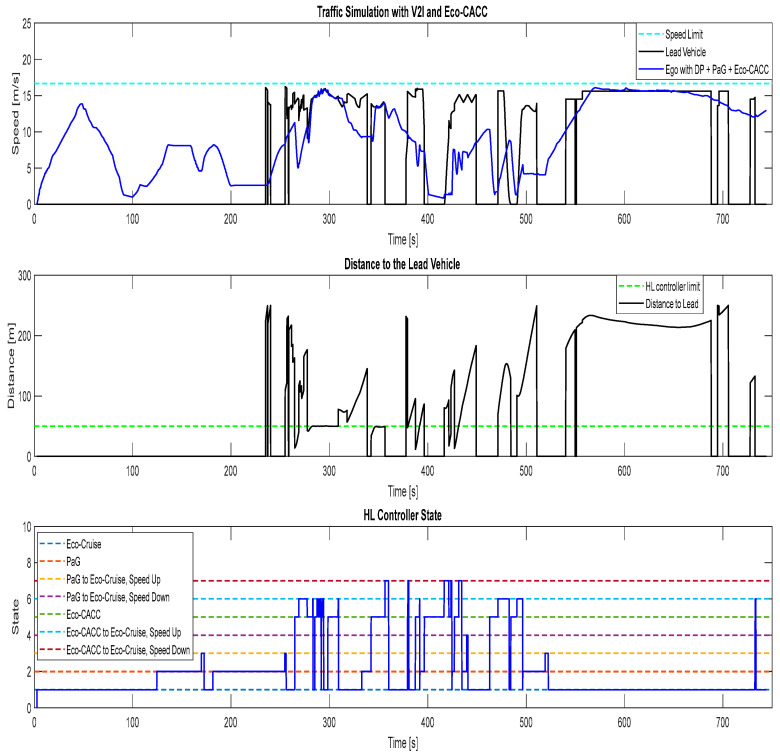
Traffic simulation for ego vehicle with V2I and Eco-CACC, case 5.

**Table 1 sensors-23-08416-t001:** Results for the fuel economy of the ego vehicle in no-traffic network.

Simulation Case Number	Simulation Scenario Name	Total Fuel Consumption (g)	% Fuel Consumption Reduction with Respect to Case #1
1	Eco-Cruise only (no traffic, vehicle stops at all traffic lights)	395.85	-
2	Eco-Cruise with Eco-Stop and PaG (no traffic, V2I only)	382.17	3.46%

**Table 2 sensors-23-08416-t002:** Results for the fuel economy of the ego vehicle in a traffic network.

Simulation Case Number	Simulation Scenario Name	Total Fuel Consumption (g)	% Fuel Consumption Reduction with Respect to Case #3
3	Eco-Cruise with Eco-Stop and PaG and ACC (V2I + no V2V)	454.20	-
4	Eco-Cruise with Eco-Stop and PaG and CACC (V2I + V2V)	447.37	1.51%
5	Eco-Cruise with Eco-Stop and PaG and Eco-CACC (V2I + V2V)	425.12	6.41%

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
