# Peer review of "A Comprehensive Eco-Driving Strategy for CAVs with Microscopic Traffic Simulation Testing Evaluation"

_sensors, 2023, doi:10.3390/s23208416_

Round 1
Reviewer 1 Report
Dear Authors,
Congratulations on the submitted paper! The paper is very interesting and addresses an actual problem consisting of the proper use of new technologies such as V2I and V2V in optimizing fuel consumption. Since the researched problem involves connected autonomous vehicles, the paper fits very well to the scope of Sensors journal, implying the real-time data acquisition of data from a sensor-based environment and the design of appropriate mechanism for real-time control.
Please see below my comments and some suggestions for improvement:
1. The methodology is well explained and contains all the formalisms that facilitate the reproducibility of this work, having a high impact on future research.
2. There are some writing aspects that should be improved: add the extended version of the abbreviated notations as their first appearance (CAV - line 26) and after that use only the abbreviations (lines 357-359 - the extended versions are already specified in lines 65 and 74), remove additional spaces (line 40), in equations (1) and (2) the terms are overwritten with the line numbers (lines 199-203).
3. The lane change is described as the action of the target follower vehicle based on the behavior of the leader vehicle, the control strategy of the follower vehicle implying a lane change action to ensure safety movement (lines 183-189). How do the proposed algorithms behave in the case of a lane change from the perspective of leader vehicle change (the leader vehicle changes the lane or a new vehicle from adjacent traffic lanes joins the target lane, and the target vehicle should adapt the control strategy based on this new leader?).
4. Is there a specific package from MATLAB Simulink that should be installed for the simulation experiments? Give these details to ensure the reproducibility of the simulation experiment.
Author Response
October 5, 2023
Manuscript ID: sensors-2648189
Title: A Comprehensive Eco-Driving Strategy for CAVs with Microscopic Traffic Simulation Testing Evaluation
Authors: Ozgenur Kavas-Torris, Levent Guvenc *
The authors would like to thank the editor and reviewers for their valuable comments and suggestions which were very useful in improving the paper.
Note: Reviewer comments are in italic and boldface and the authors’ replies are in normal typeface. Equation and figure numbers correspond to the original version of the paper before the major revision unless stated otherwise.
Changes made are highlighted in yellow in the revised paper.
REPLY TO REVIEWER 1
Dear Authors,
Congratulations on the submitted paper! The paper is very interesting and addresses an actual problem consisting of the proper use of new technologies such as V2I and V2V in optimizing fuel consumption. Since the researched problem involves connected autonomous vehicles, the paper fits very well to the scope of Sensors journal, implying the real-time data acquisition of data from a sensor-based environment and the design of appropriate mechanism for real-time control.
The authors thank the reviewer for their comment and for reviewing our paper.
Please see below my comments and some suggestions for improvement:
- The methodology is well explained and contains all the formalisms that facilitate the reproducibility of this work, having a high impact on future research.
The authors thank the reviewer for their comment.
- There are some writing aspects that should be improved: add the extended version of the abbreviated notations as their first appearance (CAV - line 26) and after that use only the abbreviations (lines 357-359 - the extended versions are already specified in lines 65 and 74), remove additional spaces (line 40), in equations (1) and (2) the terms are overwritten with the line numbers (lines 199-203).
The authors thank the reviewer for their comment. The following text was added to the revised paper (also highlighted in yellow in the revised version of the paper):
Connected and Autonomous Vehicle (CAV) (CAV - line 26)
technology. Altan (space was removed in line 41 in the revised version)
Equation 1 and 2 were reformatted not to be overwritten the line numbers (line 203 to 208)
The ACC model (line 362 in the revised version)
The CACC (line 364 in the revised version)
- The lane change is described as the action of the target follower vehicle based on the behavior of the leader vehicle, the control strategy of the follower vehicle implying a lane change action to ensure safety movement (lines 183-189). How do the proposed algorithms behave in the case of a lane change from the perspective of leader vehicle change (the leader vehicle changes the lane or a new vehicle from adjacent traffic lanes joins the target lane, and the target vehicle should adapt the control strategy based on this new leader?).
The authors thank the reviewer for their comment. The following text was added to the revised paper:
The main goal of Lane Change in Eco-Driving of a CAV is to make sure the ego CAV can maintain the optimal Eco-Cruise speed to get maximum fuel savings, while also ensuring the safety of the ego vehicle and other nearby vehicles around adjacent lanes. If the leader vehicle changes lanes, it is not a leader vehicle anymore and the ego vehicle will revert back to Eco-Cruise until a new leader vehicle is encountered. If the Lane Change mode commanded a lane change for the ego vehicle, but a new vehicle from adjacent traffic lanes joined the target lane, then the ego vehicle would either go back to the Eco-Cruise or Car-Following modes, depending on the speed of this new vehicle in front.
- Is there a specific package from MATLAB Simulink that should be installed for the simulation experiments? Give these details to ensure the reproducibility of the simulation experiment.
The authors thank the reviewer for their comment. The following text was added to the revised paper (also highlighted in yellow in the revised version of the paper):
Details about setting up a COM connection between Simulink and Vissim can be found in [47]. Other than the COM interface between Simulink and Vissim, there was no specific Matlab Simulink package that was installed for the simulation experiments. (Line 412 in the revised version of the paper)

Reviewer 2 Report
A comprehensive Eco-Driving strategy for CAVs is presented in the work. This Eco-Driving deterministic controller for an ego CAV was equipped with Vehicle-to-Infrastructure (V2I) and Vehicle-to-Vehicle (V2V) algorithms. Based on comprehensive results, the HL controller ensures significant fuel economy improvement as compared to baseline driving modes with no collisions between the ego CAV and traffic vehicles while the driving mode of the ego CAV was set correctly under changing constraints. Overall, the paper is well written and the work is of much interest. I only have some minor comments. Please check them below:
- The abstract can be improved to some extent. Please focus on what has been done in this paper.
- Figures 5, 6, and 7 can be improved. The upper part of the start module is cut.
- To obtain the real-world behavior when the CAVs are deployed, the datasets dedicated to CAVs are needed. The paper in :an automated driving systems data acquisition and analytics platform; has presented a dedicated dataset for analyzing the CAVs behavior. Please discuss it in the paper.
-Please provide a short section to elaborate on the shortcoming of this work and the potential future work.
Author Response
October 5, 2023
Manuscript ID: sensors-2648189
Title: A Comprehensive Eco-Driving Strategy for CAVs with Microscopic Traffic Simulation Testing Evaluation
Authors: Ozgenur Kavas-Torris, Levent Guvenc *
The authors would like to thank the editor and reviewers for their valuable comments and suggestions which were very useful in improving the paper.
Note: Reviewer comments are in italic and boldface and the authors’ replies are in normal typeface. Equation and figure numbers correspond to the original version of the paper before the major revision unless stated otherwise.
Changes made are highlighted in yellow in the revised paper.
REPLY TO REVIEWER 2
A comprehensive Eco-Driving strategy for CAVs is presented in the work. This Eco-Driving deterministic controller for an ego CAV was equipped with Vehicle-to-Infrastructure (V2I) and Vehicle-to-Vehicle (V2V) algorithms. Based on comprehensive results, the HL controller ensures significant fuel economy improvement as compared to baseline driving modes with no collisions between the ego CAV and traffic vehicles while the driving mode of the ego CAV was set correctly under changing constraints. Overall, the paper is well written and the work is of much interest. I only have some minor comments. Please check them below:
The authors thank the reviewer for their comment and for reviewing our paper.
- The abstract can be improved to some extent. Please focus on what has been done in this paper.
The authors thank the reviewer for their comment. The following text was added to the Abstract section of the revised paper: (also highlighted in yellow in the revised paper.)
In this paper, a comprehensive deterministic Eco-Driving strategy for Connected and Autonomous Vehicles (CAVs) is presented. In this setup, multiple driving modes calculate speed profiles ideal for their own set of constraints simultaneously to save fuel as much as possible, while a High Level (HL) controller ensures smooth and safe transitions between the driving modes for Eco-Driving. This Eco-Driving deterministic controller for an ego CAV was equipped with Vehicle-to-Infrastructure (V2I) and Vehicle-to-Vehicle (V2V) algorithms. This comprehensive Eco-Driving strategy and its individual components were tested using simulations to quantify the fuel economy performance. Simulation results are used to show that the HL controller ensures significant fuel economy improvement as compared to baseline driving modes with no collisions between the ego CAV and traffic vehicles, while the driving mode of the ego CAV was set correctly under changing constraints. For the microscopic traffic simulations, a 6.41% fuel economy improvement was observed for the CAV that was controlled by this comprehensive Eco-Driving strategy.
- Figures 5, 6, and 7 can be improved. The upper part of the start module is cut.
The authors thank the reviewer for their comment. The following figures were added to the revised paper: (also highlighted in yellow in the revised paper.)
Figure 5
Figure 6
Figure 7
- To obtain the real-world behavior when the CAVs are deployed, the datasets dedicated to CAVs are needed. The paper in :an automated driving systems data acquisition and analytics platform; has presented a dedicated dataset for analyzing the CAVs behavior. Please discuss it in the paper.
The authors thank the reviewer for their comment. The following text and reference were added to the Conclusions section of the revised paper: (also highlighted in yellow in the revised paper.)
To obtain real-world behavior when the CAVs are deployed, datasets dedicated to CAVs are needed. This has not been treated in the current paper but there are such papers in the literature where such data is collected. For example, the paper in [49] has presented a dedicated dataset for analyzing the CAV’s behavior.
[49] Xin Xia, Zonglin Meng, Xu Han, Hanzhao Li, Takahiro Tsukiji, Runsheng Xu, Zhaoliang Zheng, Jiaqi Ma, “An automated driving systems data acquisition and analytics platform,” Transportation Research Part C: Emerging Technologies, Volume 151, 2023, 104120.
-Please provide a short section to elaborate on the shortcoming of this work and the potential future work.
The authors thank the reviewer for their comment. The following text was added to the Conclusions and Future Work section of the revised paper: (also highlighted in yellow in the revised paper.)
For future work, different driving modes that were presented here can be combined as part of an MPC with varying constraints under different driving conditions to improve the complete Eco-Driving strategy of a CAV presented in this paper.
There is also potential for improvement for the High Level (HL) controller. In the simulation results, it was seen that for some cases during car following, the HL controller switched between different driving modes very rapidly. In real life implementation, this rapid switching between driving modes would diminish the ride comfort for the passengers. To eliminate this rapid switching issue in the HL controller, a dead-zone can be included in the controller. When controllers have dead-zones, they do not respond to the change in the input within the dead-zone region [48]. By exploring the addition of a dead-zone to the HL controller, the rapid switching issue might be eliminated.
Within the scope of this paper, it was assumed that the functional safety of the ego CAV was satisfied and there were no malicious agents for the V2I, V2V and V2X communication. However, in real life, there could be cyber security threats to the functional safety of a CAV due to malicious road agents. For example, malicious agents could broadcast inaccurate acceleration information to other CAVs on the roadway, or they could intentionally drive in an erratic manner. For safe and reliable real-life implementation and VIL testing, the cyber security and functional safety aspects of CAVs need to be explored further.

Reviewer 3 Report
Thank you for your diligent effort and the work invested in producing this paper. Below are my comments and suggestions to further enhance the quality and clarity of your manuscript:
• In the abstract, mention the full name of "CAV" when referring to it for the first time; thereafter, you can use the acronym.
• In the abstract, there is no numerical justification to support the author's claims or results of comparative analysis to demonstrate superior performance.
• The abstract is not detailed enough. Readers expect to see more information about the methodology, results, and conclusion in the abstract. Therefore, the abstract of this paper needs significant improvement.
• The introduction needs substantial improvement as it lacks sufficient information. For example, what problem are you attempting to solve? Are there any existing solutions? Which is the best, and why? What is the main limitation of the best and existing approaches? What do you propose to change or improve to make it better?
• In this research, the introduction section focuses solely on the literature review.
• The novelty of this research is unclear. For instance, what makes the proposed Comprehensive Eco-Driving Strategy developed in this research novel? Is there any improvement in the results compared to the literature? Has this study enhanced performance?
• Equations should be cited in the paper.
• It would be preferable to rename the "Conclusions" section to "Conclusions and Future Work" or "Conclusions and Recommendations."
• How do the authors validate their work?
• What are the experimental parameters?
Author Response
October 5, 2023
Manuscript ID: sensors-2648189
Title: A Comprehensive Eco-Driving Strategy for CAVs with Microscopic Traffic Simulation Testing Evaluation
Authors: Ozgenur Kavas-Torris, Levent Guvenc *
The authors would like to thank the editor and reviewers for their valuable comments and suggestions which were very useful in improving the paper.
Note: Reviewer comments are in italic and boldface and the authors’ replies are in normal typeface. Equation and figure numbers correspond to the original version of the paper before the major revision unless stated otherwise.
Changes made are highlighted in yellow in the revised paper.
REPLY TO REVIEWER 3
Thank you for your diligent effort and the work invested in producing this paper. Below are my comments and suggestions to further enhance the quality and clarity of your manuscript:
The authors thank the reviewer for their comment and for reviewing our paper.
- In the abstract, mention the full name of "CAV" when referring to it for the first time; thereafter, you can use the acronym.
The authors thank the reviewer for their comment. The following text was added to the Abstract section of the revised paper: (also highlighted in yellow in the revised version of the paper)
Connected and Autonomous Vehicles (CAVs) (Line 10 - 11)
- In the abstract, there is no numerical justification to support the author's claims or results of comparative analysis to demonstrate superior performance.
The authors thank the reviewer for their comment. The following text was added to the Abstract section of the revised paper: (also highlighted in yellow in the revised version of the paper)
For the microscopic traffic simulations, a 6.41% fuel economy improvement was observed for the CAV that was controlled by this comprehensive Eco-Driving strategy. (Line 20-21)
- The abstract is not detailed enough. Readers expect to see more information about the methodology, results, and conclusion in the abstract. Therefore, the abstract of this paper needs significant improvement.
The authors thank the reviewer for their comment. The following text was added to the Abstract section of the revised paper: (also highlighted in yellow in the revised version of the paper)
In this paper, a comprehensive deterministic Eco-Driving strategy for Connected and Autonomous Vehicles (CAVs) is presented. In this setup, multiple driving modes calculate speed profiles ideal for their own set of constraints simultaneously to save fuel as much as possible, while a High Level (HL) controller ensures smooth and safe transitions between the driving modes for Eco-Driving. This Eco-Driving deterministic controller for an ego CAV was equipped with Vehicle-to-Infrastructure (V2I) and Vehicle-to-Vehicle (V2V) algorithms. This comprehensive Eco-Driving strategy and its individual components were tested using simulations to quantify the fuel economy performance. Simulation results are used to show that the HL controller ensures significant fuel economy improvement as compared to baseline driving modes with no collisions between the ego CAV and traffic vehicles, while the driving mode of the ego CAV was set correctly under changing constraints. For the microscopic traffic simulations, a 6.41% fuel economy improvement was observed for the CAV that was controlled by this comprehensive Eco-Driving strategy.
- The introduction needs substantial improvement as it lacks sufficient information. For example, what problem are you attempting to solve? Are there any existing solutions? Which is the best, and why? What is the main limitation of the best and existing approaches? What do you propose to change or improve to make it better?
The authors thank the reviewer for their comment. The problem being solved is how to improve fuel efficiency of a vehicle (eco-driving) using connectivity with the infrastructure and nearby vehicles. Existing solutions were presented in the literature review part of the Introduction. The aim of the paper is to improve fuel economy which has been shown in the simulation experiment results. The following was added to the Introduction part:
The problem being addressed here is how to improve fuel efficiency of a vehicle (eco-driving) using connectivity with the infrastructure and nearby vehicles. Existing solutions are first presented in the literature review below, followed by the contributions made in this paper. The aim of this paper is to improve fuel economy which is shown in the simulation experiment results parts of the paper.
- In this research, the introduction section focuses solely on the literature review.
The authors thank the reviewer for their comment. The introduction section of the revised paper also includes the contributions of the paper.
- The novelty of this research is unclear. For instance, what makes the proposed Comprehensive Eco-Driving Strategy developed in this research novel? Is there any improvement in the results compared to the literature? Has this study enhanced performance?
The authors thank the reviewer for their comment. The highlighted section summarized the contributions of our paper.
This study shows relative fuel savings each component provides to CAVs, how each component can be improved and what constitutes the largest effect on fuel savings. It has been shown that the complete Eco-Driving architecture presented in this paper is applicable to be used in real life in actual vehicles. The main contribution of this paper is the development and simulation validation of an integrated eco-driving system that uses V2I to handle realistic situations with infrastructure (STOP signs, traffic lights) and V2V to handle interactions with other vehicles. The other contributions that help this main contribution are summarized as follows:
- V2I and V2V algorithms were developed to control the longitudinal motion of a CAV for Eco-Driving.
- The High Level (HL) controller was also tested in a traffic simulator with realistic traffic flow. The traffic vehicles were controlled by the traffic simulator, and had default car following models, which enabled them to change lanes when they were behind slower vehicles. Thus, the traffic vehicles created dynamically changing constraints on the HL controller. It was observed that the HL controller ensured that no collisions were observed between the ego CAV and traffic vehicles, and the driving mode of the ego CAV was set correctly under changing constraints.
- The High Level (HL) controller designed for the comprehensive Eco-Driving of a CAV enabled fuel savings.
- Equations should be cited in the paper.
The authors thank the reviewer for their comment. The paper was thoroughly proofread, and equations were cited in the revised paper.
- It would be preferable to rename the "Conclusions" section to "Conclusions and Future Work" or "Conclusions and Recommendations."
The authors thank the reviewer for their comment. The following text was added to the Conclusions and Future Work section of the revised paper: (also highlighted in yellow in the revised paper.)
For future work, different driving modes that were presented here can be combined as part of an MPC with varying constraints under different driving conditions to improve the complete Eco-Driving strategy of a CAV presented in this paper.
There is also potential for improvement for the High Level (HL) controller. In the simulation results, it was seen that for some cases during car following, the HL controller switched between different driving modes very rapidly. In real life implementation, this rapid switching between driving modes would diminish the ride comfort for the passengers. To eliminate this rapid switching issue in the HL controller, a dead-zone can be included in the controller. When controllers have dead-zones, they do not respond to the change in the input within the dead-zone region [48]. By exploring the addition of a dead-zone to the HL controller, the rapid switching issue might be eliminated.
Within the scope of this paper, it was assumed that the functional safety of the ego CAV was satisfied and there were no malicious agents for the V2I, V2V and V2X communication. However, in real life, there could be cyber security threats to the functional safety of a CAV due to malicious road agents. For example, malicious agents could broadcast inaccurate acceleration information to other CAVs on the roadway, or they could intentionally drive in an erratic manner. For safe and reliable real-life implementation and VIL testing, the cyber security and functional safety aspects of CAVs need to be explored further.
- How do the authors validate their work?
The authors thank the reviewer for their comment. The following text was added to the revised paper:
The validation of the proposed strategy was carried out using realistic simulations with other traffic generated by a microscopic traffic simulator.
- What are the experimental parameters?
The authors thank the reviewer for their comment. The following text was added to the Microscopic Traffic Simulation Environment section of the revised paper: (also highlighted in yellow in the revised paper.)
The traffic vehicle compositions were the same at each simulation. Additionally, the traffic simulator spawned vehicles at a common start time for each simulation, meaning that the vehicle with a specific ID entered the roadway at the same timestamp across all simulation cases. This unity ensures that the simulation results can be com-pared with each other, since the traffic vehicles that interact with the ego vehicle appear in the simulator at the same time. Additionally, the traffic light periods for each traffic light were the same across all simulations. When it comes to experimental parameters, the distance travelled by vehicles, the inter-vehicular distance between the ego and leader vehicle, vehicle speed, simulation time, distance to traffic lights and STOP signs, SPAT for traffic lights and HL controller state were recorded and analyzed for system performance.

Round 2
Reviewer 3 Report
Thank you